# Evaluation of Linamon Red Clay, Salvador Black Cinder and Kapatagan Diatomaceous Earth of the Southern Philippines

Ivyleen C. Bernardo-Arugay [1,2,*], Fel Jane A. Echavez [2], Liberty R. Lumasag [2], Jade P. Cahigao [2], Elly U. Aligno Jr. [2], Roben Victor M. Dispo [2], Sherlyn Keh D. Dionio [2], Christian Julle C. Saladaga [2], Beverly L. Bato [2], Alyssa May Rabadon Simplicio [2] and Raymond V. Rivera Virtudazo [1,2]

1 Ceramic Engineering Program, Department of Materials and Resources Engineering and Technology, College of Engineering and Technology, Mindanao State University—Iligan Institute of Technology, Iligan City 9200, Philippines
2 Research Center for Advanced Ceramics, College of Engineering and Technology, Mindanao State University—Iligan Institute of Technology, Iligan City 9200, Philippines
* Correspondence: ivyleen.arugay@g.msuiit.edu.ph

**Abstract:** The southern island of the Philippines is abundant in silicate minerals, including the province of Lanao del Norte. However, some of these resources in the region are untapped for use as raw materials in the production of various ceramic products for industrial, pharmaceutical, and nanotechnology applications. These could include tiles, sanitary ware, dinnerware, insulating bricks, porcelain, membranes and coatings. Some of the explored minerals are the red clay in the municipality of Linamon, diatomaceous earth in Kapatagan and black cinder in Salvador. It is the aim of this study that these minerals are evaluated in terms of their physical and chemical properties so that these will be used for optimum application. The properties that were determined were their specific gravities, raw and fired surface colors, Atterberg limits, particle size distribution, thermal properties, morphologies and mineralogical compositions. Pellets were formed for each raw material and fired at two temperature levels 1000 °C and 1200 °C to evaluate their physical properties. Linamon red clay has a 38.88% cumulative passing size of 150 μm, and the black cinder of Salvador and diatomaceous earth of Kapatagan have cumulative passing sizes of 96.53% and 60.12% at 150-micron sieve, respectively. The common mineral contents of the three samples are montmorillonite, quartz and andesine. Black cinder fired at 1200 °C has the darkest shade of red with a greasy quasi-submetallic luster. It attained the highest fusion coverage on the platform among the three materials, which makes it a potential supplement or replacement for feldspar in clay-based triaxial materials for ceramic production. The diatomaceous earth has the potential to be a secondary clay content source and a good source of flux for a certain temperature range. Both the red clay and diatomaceous earth were classified as plastic materials that are suitable for brick production, and the red clay is also feasible for pottery production. These are a few of the features of the clay minerals in the region that present suitable properties for application as raw materials in the production of ceramic tiles and hollow ceramic products.

**Keywords:** cinder; diatomite; red clay; Philippine silicates; plasticity of soils; montmorillonite; nontronite; quartz; andesine; valorization of clays; sustainable development; thermal procedure; characterization of ceramic materials; brick tile production; Atterberg limits; fusion of silicates; economic opportunities of clays

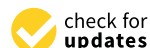



## 1. Introduction

The Philippine ceramic tile industry holds an estimated total installed capacity per annum of around thirty (30) million square meters, consisting of four major local manufacturers all stationed in the northern island of the country. One of the major challenges of the ceramic tile industry is that it only holds less than 50% of the local market share due to the

influx of imported tiles that have lower costs than the local supply. Accordingly, one of the reasons for the higher prices of local ceramic tiles than the foreign supply is the high cost and limited sources of local raw materials. The drawback of the imported tiles, however, is their non-compliance with the Philippine National Standard for ceramic tiles under PNS ISO 13006:2007, which applies to both extruded tiles and dry-pressed tiles [1]. It is therefore a great advantage for the local manufacturers to produce international quality standard ceramic tiles that are manufactured mostly from locally available resources and priced at competitive costs with foreign products.

Mindanao, the second largest island located in the southern part of the Philippine archipelago, is endowed with clay minerals. Most of these resources, including the abundant red clays in the region of Lanao del Norte, are untapped for optimum use in various ceramic applications, thus making them unproductive and low in value. The proper utilization of these available resources will not only stimulate the economy of the region and add value to vast idle silicates but will greatly help in minimizing if not eliminating the problem of the importation of raw materials, which was greatly challenged during the COVID-19 pandemic. An investigation into the properties of the clay minerals [2] in the region will lead to the optimum use and application of these resources in ceramic tiles and other products [3–5]. Establishing more manufacturers of ceramic tiles in the country, particularly in the rural regions, and using mostly the available local clay minerals will support the achievement of some of the Sustainable Development Goals of the United Nations, i.e., end poverty, promote decent work and economic growth and build sustainable cities and communities [3,6].

The use of clay in the production of different types of tiles depends on technological requirements as well as appearance, such as its color after firing and the behavior exhibited during the manufacturing process. A study [7] was conducted on the use of locally sourced raw materials in Uganda for the production of porcelain tiles. Various proportions of the materials were made ranging from 40%–60% clay, 30%–40% feldspar and 10%–30% sand and were then fired at temperatures ranging from 1050 to 1250 °C. Porcelain tiles are known to have high flexural strength greater than $35 \pm 2$ MPa and low water absorption of less than 0.5%. Based on the results, the best properties of the porcelain tile were achieved at 1200 °C for samples made of 30%–40% kaolin, 30%–40% feldspar, 20% ball clay and 10% sand with a flexural strength of 33 MPa and water absorption of 0.08% [7]. Another study on the sintering behavior of the diatomite powder bodies shows that diatomite monoliths with high porosity and compressive strength can be prepared at 1000 °C. Above 1200 °C, a melt phase forms and annihilates the intrinsic pores of the diatomite, which increases its density. Microstructural analyses reveal that the impurities in diatomite such as $Na_2O$, $K_2O$, $Al_2O_3$, CaO and MgO favor the formation of low temperature eutectics and thus forming a melt phase in the silica rich grains [8]. It is therefore important that the mineralogical, chemical, granulometric and rheological characteristics of the clays be determined to assess the potential industrial application of the raw materials [3,9,10]. One of the abundant clay minerals in the region of Lanao del Norte is the diatomaceous earth in Kapatagan, which is being quarried for use as filling materials.

The aim of this study is to evaluate the properties of some of the abundant clay minerals found in the region, which are the red clay in the municipality of Linamon, diatomaceous earth in Kapatagan and black cinder in Salvador, Lanao del Norte. The sources of these minerals are estimated to have land areas of about 38,000 square meters in the case of Linamon clay and 32,000 square meters in the case of Salvador black cinder. However, no information has been provided by the land owner of the deposit for Kapatagan diatomaceous earth. The physical properties, plasticity, mineralogy, chemical compositions, thermal behavior, morphology and sintering behavior of these clay minerals will be evaluated to determine their viability as raw materials in the making of ceramic industrial products. Utilizing these abundant resources will ensure cheaper raw materials than imported supplies, which could promote investment for an expansion or establishment of a new ceramic manufacturing industry in the region. This would then lead to the generation of more employment in

businesses with direct and indirect linkages with the ceramic industries and later foster a more sustainable community.

## 2. Materials and Methods

### 2.1. Sample Preparation of the Raw Materials

Sample collection was conducted in three different localities for the various raw silicates, namely Linamon for red clay (LRC), Salvador for black cinder (SBC) and Kapatagan for diatomaceous earth (KDE), as shown in Figure 1. Sampling was then performed to collect representative samples from each bulk of raw material using the coning and quartering method. The samples were dried to eliminate moisture and then crushed using a UA V-Belt Drive pulverizer (BICO Braun International, Burbank, CA, USA) with a power of 3 HP, 8-inch grinding plates, and a speed of 900 rpm. Using a 100-mesh screen, the pulverized samples were dry screened to collect samples of sizes less than 150 microns.

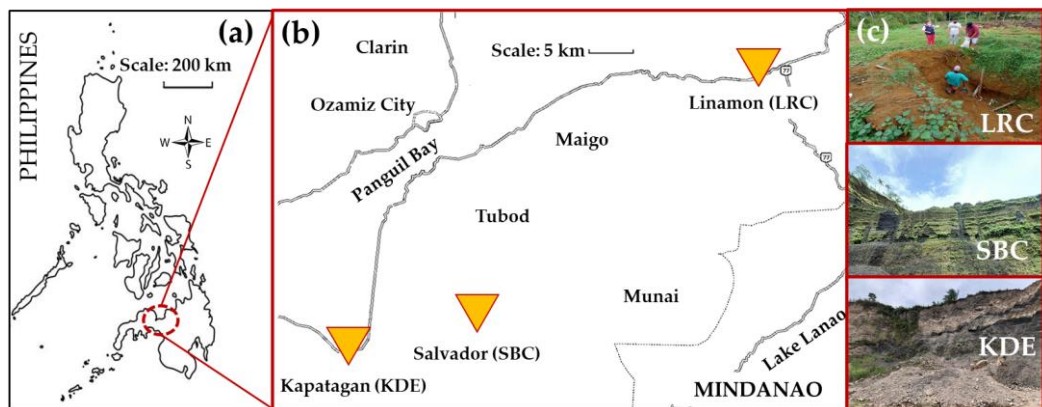

**Figure 1.** Various locations of the three raw silicates located in (**a**) Mindanao, Philippines. (**b**) The three municipalities where the raw materials are sourced on the island of Mindanao (source: Google Maps). (**c**) Actual sites (images) of the raw materials: LRC in Linamon, SBC in Salvador and KDE in Kapatagan.

### 2.2. Raw Material Characterization

A particle size distribution (PSD) analysis of each raw material was performed using a Tyler standard sieve series ranging from 4 to 200 mesh (4.75 to 0.075 mm) in a mechanical sieve shaker in accordance with ASTM D6913-04 [11]. The specific gravity of the representative samples with a particle size of less than 150 μm was determined using the pycnometer method as per ASTM D854-14 [12]. The liquid limit, plastic limit and plasticity index of the samples were evaluated to determine their plastic behavior in accordance with ASTM D4318-10 [13]. Chemical analysis was also performed using an Olympus Innov-X Delta X Pro X-ray Fluorescence Spectrometer (Olympus Innov S, Woburn, MA, USA). The mineralogical phases were identified from X-ray diffraction (InXitu BTX II XRD Analyzer, InXitu, CA, USA), with CuKα radiation (1.541838 Å), 40 kV voltage and 30 mA current. The results of the X-ray Powder Diffraction of each raw material were primarily based on "figure-of-merit" values from available references [14–18]. Samples with a particle size of less than 150 microns, weighing approximately 50 to 115 milligrams, were subjected to thermal characterization using a Shimadzu DTG-60H thermogravimetric analyzer (Shimadzu Corporation, Kyoto, Japan). The sample was placed in an alumina crucible and heated to 1000 °C at a heating rate of 10 °C/min under air environment. Another set of 150-micron samples were prepared for morphological analysis using a JEOL JSM-6510 Series Scanning Electron Microscope (SEM, JEOL Ltd., Tokyo, Japan). Prior to the SEM analysis, the samples were sputter coated with platinum at 40 mA current for 40 s.

### 2.3. Specimen Preparation for Fusion Test and Sintering Treatments

The fusion test or "button test" was conducted on the powders of the raw materials passing a 100-mesh sieve (150 μm). This test provides a qualitative evaluation of the raw materials pertaining to the character of fusibility and fired color of the material over a wide range of temperatures [19,20]. Furthermore, it is common laboratory practice that fusion tests be used to provide a qualitative evaluation of the luster, opacity/translucency, surface texture and integrity of shape of the pellet fired at a certain range of temperatures. Distilled water was used as a binder at 5 wt% to make a plastic mass of the powders. The plastic mass was mixed until it became a homogenous mixture. Pellets were formed from the homogenized plastic mass using a pellet press (PARR Instrument Company, Moline, IL, USA) with a diameter of 12.92 mm and a height of 13.21 ± 0.35 mm. The mass, height and diameter of each pellet were measured for the determination of its shrinkage. Flat tiles with a dimension of 25 mm × 25 mm × 9 mm of formulated ceramic powders were prepared to be used as platforms for the pellets during heat treatment. These pellets were fired separately at two different temperatures, 1000 °C and 1200 °C, in an electric muffle furnace (SH Scientific, SH-FU-36MHSH Scientific Co., Ltd., Sejong, Korea). After firing, the equipment was switched off and the samples were furnace-cooled. The physical characteristics of the fired pellets were determined in terms of total linear shrinkage, water absorption and apparent porosity. The equations used to obtain the aforementioned properties can be found in articles that have already been published [7,10,19–22].

## 3. Results and Discussion

### 3.1. Raw Material Characterization

#### 3.1.1. Particle Size Distribution and Specific Gravity

The particle size distributions of the raw materials are presented in Figure 2. The particle size of LRC has a median diameter D50 of 0.172 mm (172 μm) and has a 38.88% cumulative passing size of 0.150 mm (150 μm). The particle sizes of SBC and KDE have median diameters of 0.109 mm (109 μm) and 0.140 mm (140 μm), respectively. Moreover, SBC and KDE have cumulative passing sizes of 96.53% and 60.12% at 150 μm sieve, respectively, with SBC having the highest value. The specific gravity of the raw materials is presented in Table 1, where KDE has the lowest value among the three samples.

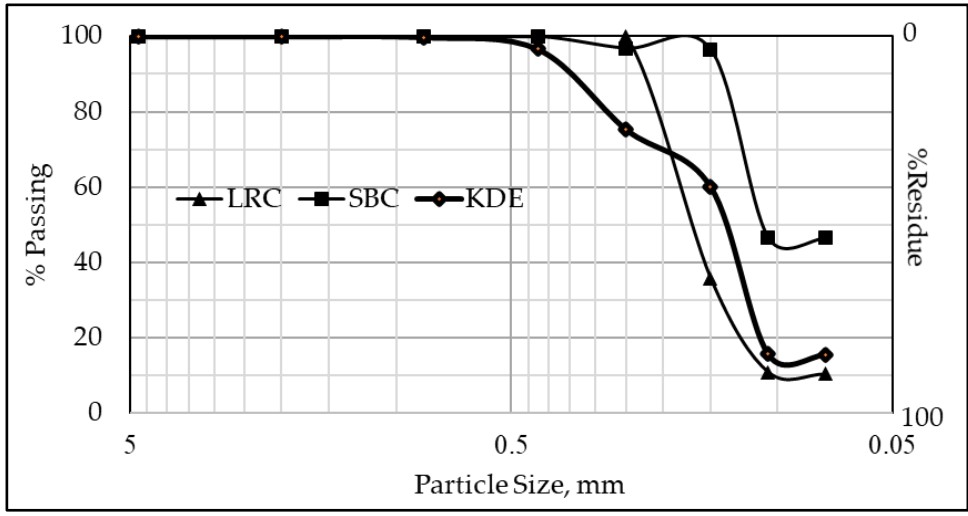

**Figure 2.** Particle size distribution (PSD) of raw materials.

**Table 1.** Specific gravity of raw materials.

| Raw Materials | Specific Gravity |
|---|---|
| LRC | $2.38 \pm 0.05$ |
| SBC | $2.63 \pm 0.07$ |
| KDE | $1.95 \pm 0.25$ |

### 3.1.2. Plasticity

The liquid limit, plastic limit and plasticity index of the powders of the three raw materials after being passed through a 425-micron sieve were investigated. The Atterberg limits of LRC shows a liquid limit of 62% and a plastic limit of 40%, which resulted in a plasticity index of 22%. SBC is reported to be nonplastic in accordance with ASTM D4318-10 [13] since neither the liquid limit nor the plastic limit could be determined. Adding more water to form the plastic mass still resulted in the crumbling of the samples as shown in Figure 3. The Atterberg limits of the KDE shows a liquid limit of 79% and a plastic limit of 61%, which resulted in a plasticity index of 18%. Accordingly, clays that present a plasticity index between 20% and 35% are denominated high plasticity clays and those with plasticity index values between 10% and 20% are denominated medium plasticity clays. Following these criteria, LRC can be classified as high plasticity clay, coinciding with the appearance of smectites, as presented in the XRD results where samples relatively high in smectite and illite are the most plastic [23] while KDE is classified as a medium plasticity clay. Figure 4 shows a plot of the plasticity of LRC and KDE. Both LRC and KDE were categorized as elastic silt in accordance with ASTM D2487 [24], meaning that elastic silt has a liquid limit of 50 or greater. LRC exhibited a lower liquid limit and plastic limit than previous studies [5] on nickel laterite mine waste (NMW) and expansive soils [25,26]. Furthermore, Figure 5 shows an extrusion prognostic using the plasticity of the samples studied by the Atterberg limits [27]. The plastic limit is related to the amount of moisture required for clay to reach a consistency that makes the extrusion process possible while the plasticity index falls between a plastic and sludge consistency range [27]. The plasticity index must be more than 10% for the clays to be utilized in the production of bricks via an extrusion process [27,28]. In this study, both LRC and KDE have a plasticity index higher than 10%. However, it can be observed that both LRC and KDE are located outside the limits of the acceptable extrusion region, indicating that these samples will result in higher linear shrinkage, which is prone to cause problems when forming products. This property may be improved via the addition of nonplastic materials such as feldspar and silica for usability in the ceramic industry. Figure 5 shows that both LRC and KDE are suitable for brick production using plastic mass forming or hand wedging [27,29]. LRC is likely acceptable only for pottery production [27]. The unique Atterberg limit values of the three raw materials could be attributed to their varying population of fine particle sizes, particularly in the fraction of particles with diameters of less than 2 microns, which is the most important driver of clay mineral flexibility [30]. In addition to the fine fraction of particles, the quantity and type of clay minerals, as well as the type of absorbed cations [31], may also affect the Atterberg limit values.

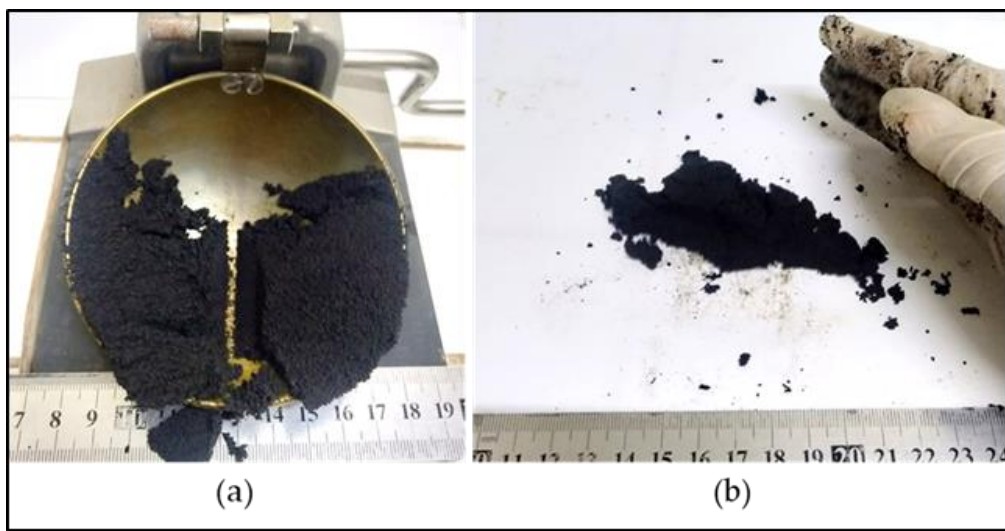

**Figure 3.** Failure in the determination of the Atterberg limits of SBC. (**a**) Liquid limit. (**b**) Plastic limit.

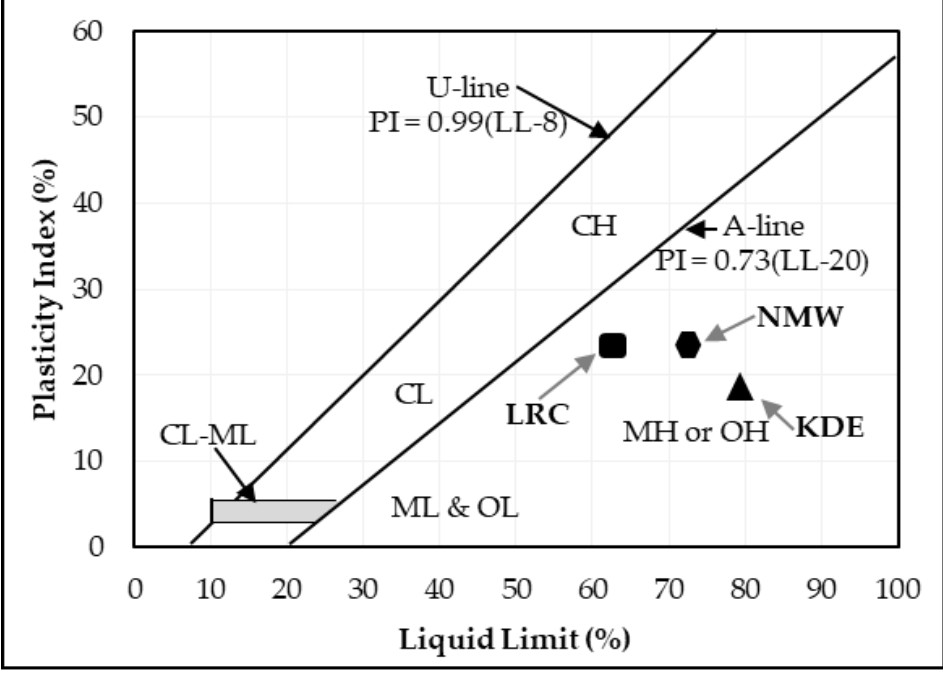

**Figure 4.** Plasticity index and liquid limit of LRC and KDE on the plasticity chart [5,32].

### 3.1.3. Mineralogical and Chemical Analysis

The chemical composition of the three raw materials are shown in Table 2. They primarily consist of silica ($SiO_2$), alumina ($Al_2O_3$), iron oxide ($Fe_2O_3$), magnesia (MgO) and titania ($TiO_2$). Trace elements in the three samples include oxides of manganese (MnO), nickel (NiO) and chromium ($Cr_2O_3$). Both SBC and KDE contain oxides of calcium (CaO), which is not present in LRC. LRC has relatively high alumina and iron oxide content at 34.39% and 14.37%, respectively. This amount of iron oxide in the LRC is lower than the usual red clays used for traditional ceramic production [4,5] and is still lower than the iron oxide of the NMW of Agusan del Norte [5,33], which was developed for tile and terracotta production [34]. The amount of alumina and iron oxide content in LRC is higher than in expansive soils [25,26], but the silica content of LRC is lower than that of expansive soils. Among the three samples, only LRC is considered a high alumina clay deposit since its alumina content is within the range 25%–35% [35]. More than 50% of the composition of

SBC and KDE is silica ($SiO_2$), while LRC has a lesser content of 42.63%. Both KDE and SBC are natural pozzolanic materials or silicate glass materials according to ASTM C618-14 [36] since their total $SiO_2$, $Fe_2O_3$ and $Al_2O_3$ content is over 70% and they have a CaO content of less than 10%.

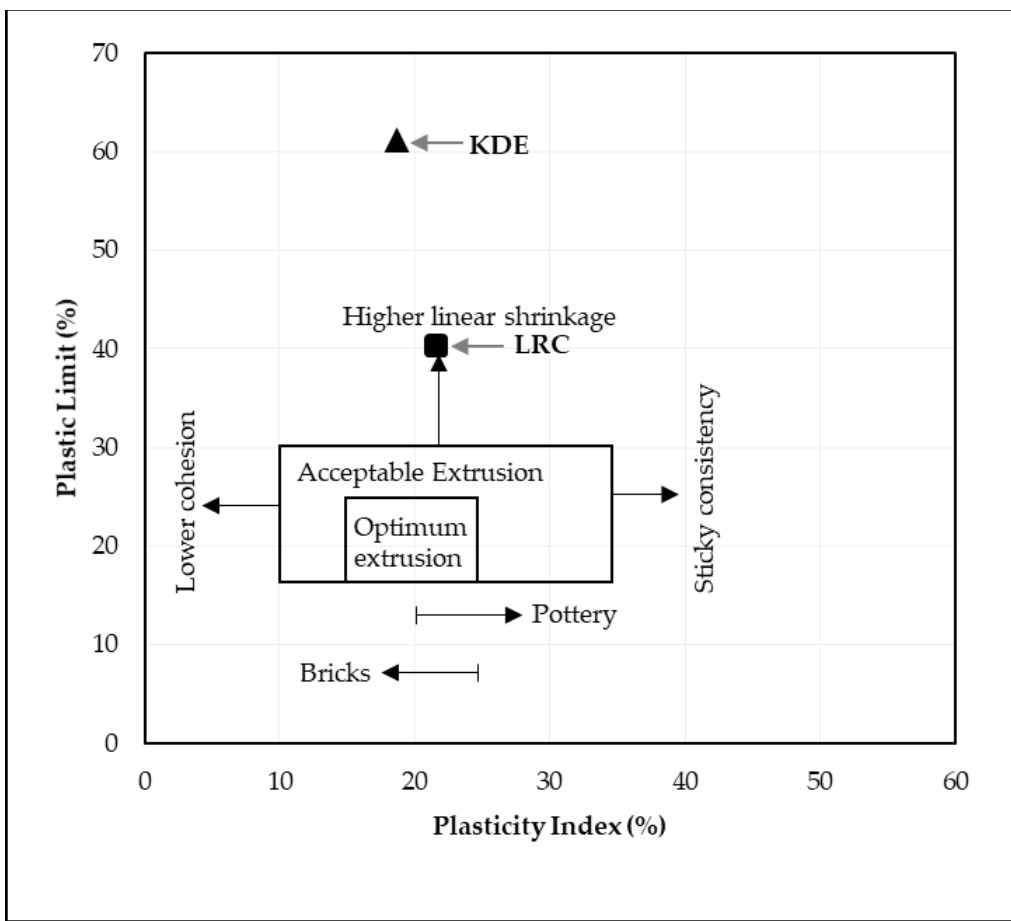

**Figure 5.** Extrusion prognostic through Atterberg limits of samples studied [27].

**Table 2.** Chemical composition of the raw materials.

| Mass % | $SiO_2$ | $Al_2O_3$ | $Fe_2O_3$ | $K_2O$ | MgO | CaO | NiO | $Cr_2O_3$ | MnO | $TiO_2$ |
|---|---|---|---|---|---|---|---|---|---|---|
| LRC | 42.63 | 34.39 | 14.37 | — | 6.1 | — | 0.03 | 0.1 | 0.2 | 2.19 |
| SBC | 55.68 | 17.15 | 6.42 | — | 9.75 | 8.61 | 0.02 | 0.04 | 0.1 | 2.23 |
| KDE | 58.16 | 18.95 | 8.97 | — | 7.83 | 4.85 | 0.01 | 0.01 | 0.07 | 1.15 |
| NMW [5] | 24.34 | 9.20 | 46.26 | — | 15.10 | 0.71 | 1.47 | 2.14 | 0.72 | 0.05 |

Note: "—" means below detection limit.

The XRD patterns of each raw material are shown in Figure 6. LRC contains minerals such as montmorillonite, nontronite, lizardite, forsterite, halloysite, goethite, hematite, anorthite, andesine and quartz. SBC has various minerals identified with dominant contents of andesine, anorthite, augite, montmorillonite, illite, quartz and lizardite. The most substantial peak intensities in this pattern were a feldspar group, which is a typical XRD pattern for volcanic ash and volcanic tuff as referenced in other studies [37–40]. KDE contains minerals such as kaolinite, halloysite, montmorillonite, illite, saponite, andesine, nontronite, quartz and hematite, as shown in Figure 6c. Few of these minerals are also present in other diatomaceous earth samples [24,41]. Moreover, these phyllosilicate minerals, such as montmorillonite, nontronite and lizardite, may potentially be useful for the slip casting production of ceramic tiles [4,5,42–45]. There were few carbonates suspected on the three

samples; however, the results of the TGA-DTA of the samples exhibited otherwise and will be discussed further in the Thermal Analysis section. Furthermore, proximate analyses of the samples were conducted to determine the clay content and free phase silica using the feldspar convention [19] based on chemical and mineralogical analyses. LRC was found to have the highest clay content (~77.56%), followed by KDE with a value of 24.70%. SBC had the lowest clay content (~3.68%), which is classified as a non-plastic material. This is consistent with the behavior of SBC during the determination of its Atterberg limits as shown in Figure 3. SBC was found to contain the highest free phase silica content of 35.32%, followed by KDE with 34.63%. LRC had the lowest content of free phase silica at 1.93%. Based on the XRD of SBC (Figure 6b), the oxide of calcium could be attributed to montmorillonite, andesine, augite, and anorthite. Meanwhile, the oxide of calcium for KDE could be attributed to saponite, montmorillonite and andesine as shown in Figure 6c.

### 3.1.4. Morphological Analysis

The morphological analyses of the three raw materials show varying structures as seen in the SEM images in Figure 7. The structure of the LRC is relatively loose and has irregular-shaped particles with rough and flaky edges. The SBC shows glass fragments with blocky and platelike shapes as illustrated in Figure 7b. The presence of leaf-shaped or filamentous particles was also seen in the SBC sample, which could have resulted in the occurrences of micro-intergranular pores. Moreover, KDE was found to contain the anticipated diatoms showing tangled porous particles and capsule-shaped frustules, as shown in Figure 7c, which are typical structures of diatomaceous earth deposits [41,46–48]. This structure causes the low specific gravity of KDE, which is consistent with the obtained result as shown in Table 1. It is also evident that a platelike structure is found in the KDE sample, most likely due to the presence of phyllosilicate, as shown in Figure 6c.

### 3.1.5. Thermal Analysis

The DTA curves and mass losses obtained from the thermogravimetric analysis of each raw material are shown in Figure 8. It can be seen that the total mass loss of LRC, SBC and KDE are 20.49%, 0.88% and 12.84%, respectively. The DTA curves of the raw materials lead to the following results: significant endothermic peak at 97.62 °C for LRC, 110 °C for SBC and 97.27 °C for KDE, which can be attributed to the elimination of free water on the particle surfaces of the raw materials [19,21,41,46,49]. An endothermic band centered at ~287.67 °C is observed only in LRC, which is due to the pre-dehydration of clay minerals [21]. In addition, an endothermic peak at 525.37 °C for LRC and ~485.12 °C for KDE is due to clay minerals' dehydroxylation causing the structure of clay minerals to collapse [19,21,41,46,49]. The exothermic peak at ~937.72 °C for LRC is attributed to the decomposition of montmorillonite and formation of defects, aluminum-silicon spinel ($Si_3Al_4O_{12}$) structures (structural re-organization) and some amorphous phases [50–52]. Here, the iron ions caused the diffusion and breakage of the clay structure [53,54]. Furthermore, the various phases of decomposition that were supposed to be contributed by the suspected carbonates such as calcite [55,56], siderite [57], ankerite [57], huntite [58], dolomite [59,60] and rhodochrosite [61] are not evident in any of the three samples. This further means that the endothermic reaction around 525.34 °C in LRC in particular is only attributed to the dehydroxylation of clay minerals [19,21,41,46,49].

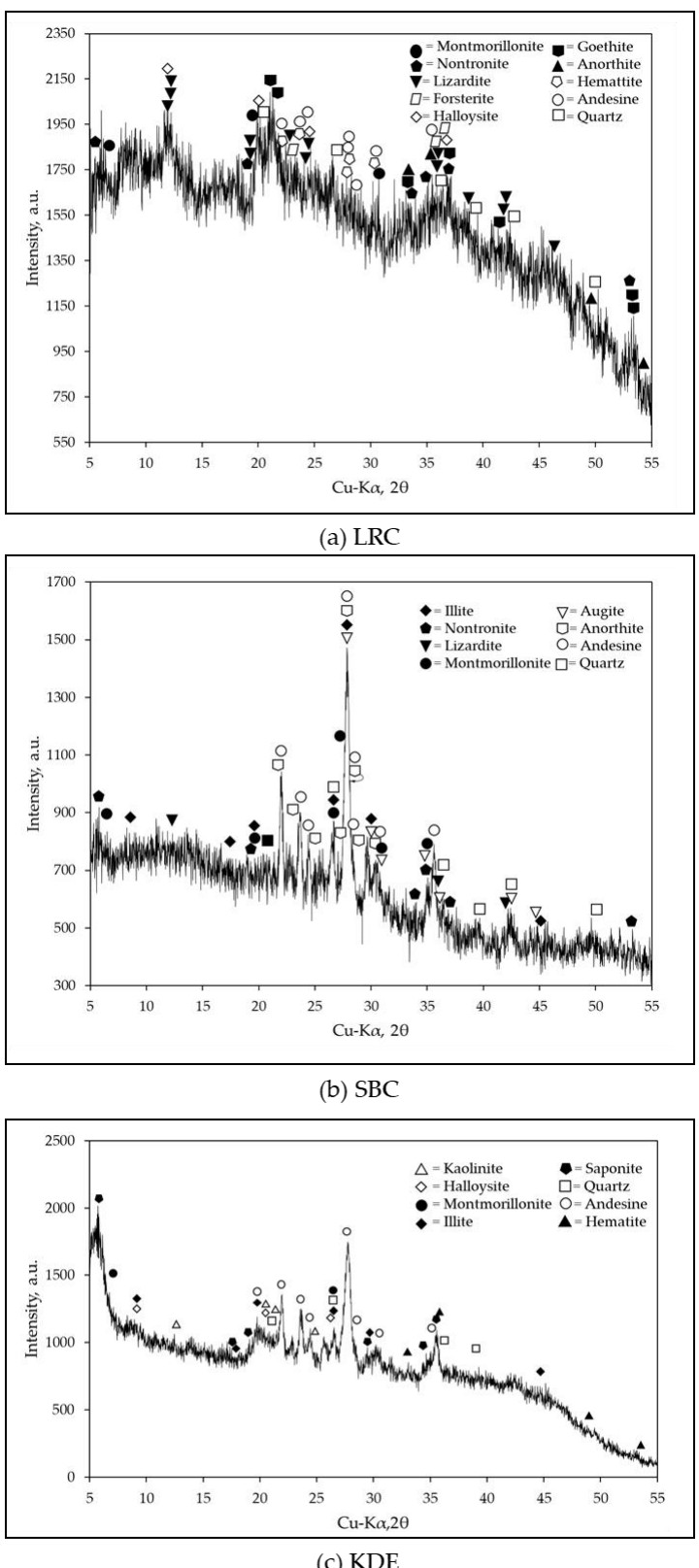

**Figure 6.** XRD patterns of raw materials: (**a**) LRC; (**b**) SBC; (**c**) KDE.

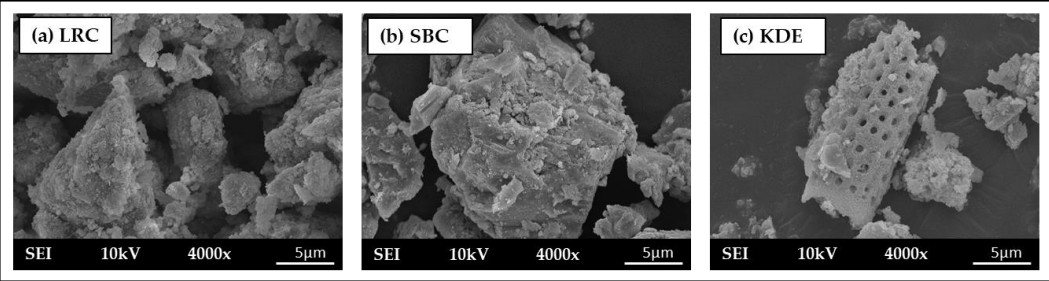

**Figure 7.** SEM images at 4500× magnification: (**a**) LRC; (**b**) SBC; (**c**) KDE.

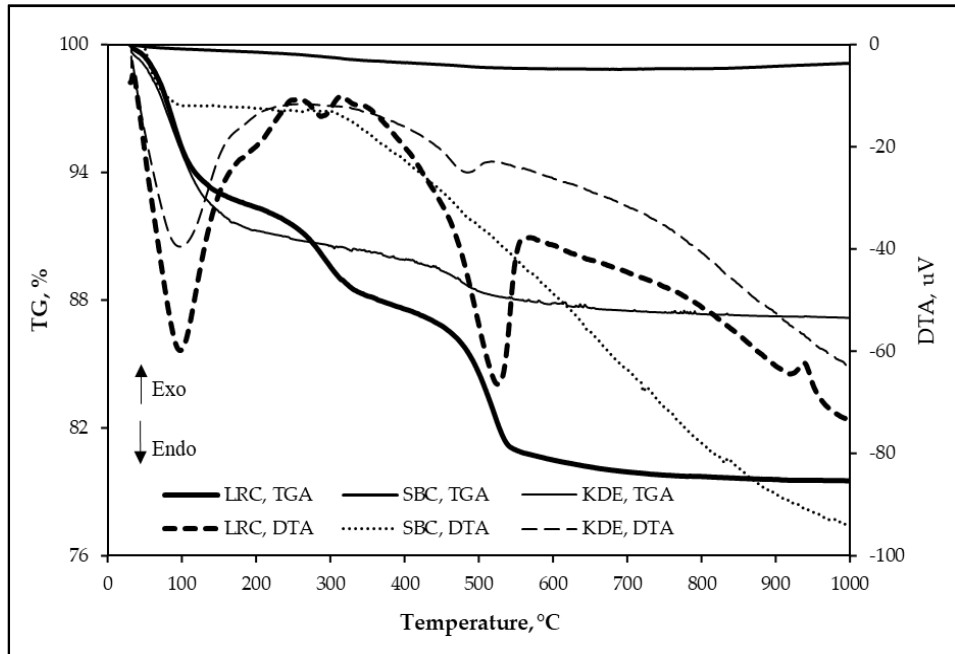

**Figure 8.** TGA-DTA of raw materials (sample: 150-micron powders of samples, crucible: alumina, heating rate: 10 °C/min, atmosphere: air).

*3.2. Fusion Test*

Images of the pellets of LRC, SBC and KDE before and after firing are shown in Figure 9, and the color of these samples were evaluated using the Munsell soil color chart [62] as seen in Table 3. The unfired pellet of LRC has a brownish yellow color, while the color of SBC is black and that of KDE is light brown. After firing, pellets of the three samples exhibited various shades of red. This is mainly due to the presence of iron oxide, where content greater than 4% ($Fe_2O_3$ plus $TiO_2$) gives an increasingly red coloration [19,48]. SBC is found to be the darkest shade of red, followed by LRC and KDE. We noted that the $Fe_2O_3/TiO_2$ ratio increases in this order, with SBC having a 2.8789 value and LRC and KDE having values of 6.5616 and 7.800, respectively. The samples exhibited a non-metallic luster after firing at 1000 °C. The results of the color of the raw materials fired at 1000 °C could predict the desired fired color for terracotta and earthenware ceramicware applications. The pellets of LRC, SBC and KDE fired at 1000 °C were not fused on flat tiles, as shown in Figure 8b.

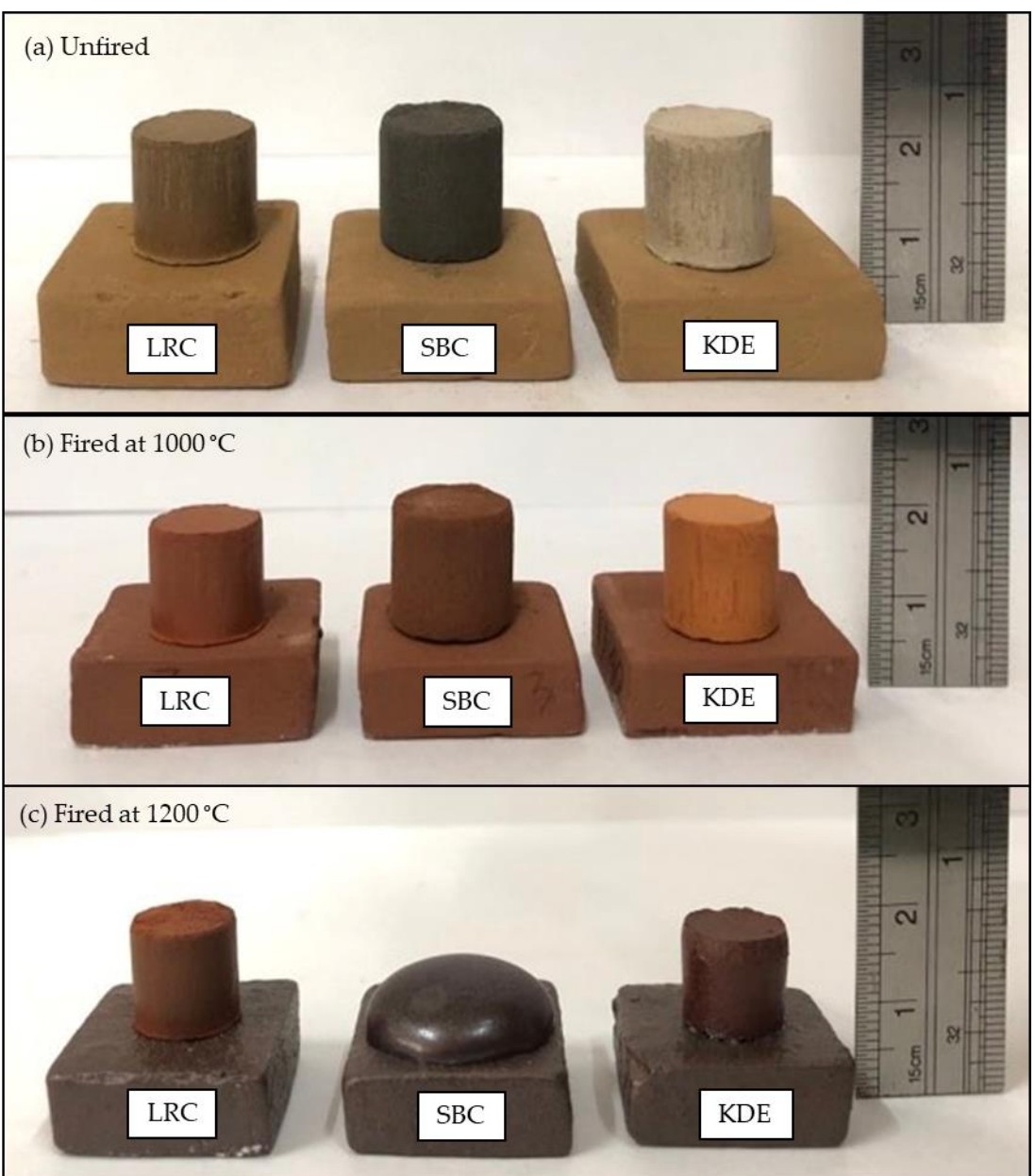

**Figure 9.** Fusion test of various raw materials. (**a**) Unfired. (**b**) Fired at 1000 °C. (**c**) Fired at 1200 °C.

**Table 3.** Evaluation of the color of samples before and after firing.

| Samples | Unfired | Fired at 1000 °C | Fired at 1200 °C |
|---|---|---|---|
| LRC | 10YR 6/8 —brownish yellow | 2.5YR 6/8—light red | 2.5YR 5/6—red |
| SBC | 10YR 2/1—black | 2.5YR 4/8—red | 2.5YR 3/6—dark red |
| KDE | 7.5YR 6/4— light brown | 5YR 6/8—reddish yellow | 2.5YR 4/8—red |

When the pellets were subjected to 1200 °C, the fused character as well as color of the pellets changed. SBC is found to be the darkest red, followed by KDE and LRC, respectively. These colors can predict the desired fired color for stoneware and terracotta applications. LRC exhibited a non-metallic luster where it is earthy or dull. Both SBC and KDE exhibited a shiny opaque luster. SBC has a greasy quasi-submetallic luster, and KDE shows a waxy luster.

At this higher firing temperature (1200 °C), LRC pellets could still be easily removed from the platform. However, both KDE and SBC were fused on the flat tile platform

as seen in Figure 9c. SBC was found to have the highest contact of fused parts on the tile, followed by KDE. This is consistent with the higher ratio of flux (MgO and CaO) to alumina of SBC (1.0706) and KDE (0.6691) compared to LRC (0.1774), as shown in Table 2. This could mean that LRC is the most refractory of the three samples, which resulted in limited fused parts on its tile platform. The fusion character of SBC and KDE could be attributed to the presence of alkaline-earth oxides acting as fluxes [19,48], as shown in Table 2, which favors the formation of low temperature eutectics and thus the formation of a melt phase that infiltrates the open pores of the structure, causing densification via liquid state sintering [8,63,64]. Hence, no data were further obtained for the physical properties of the pellets of SBC and KDE at 1200 °C.

The average results for the physical parameters of the pellets studied at different temperatures are shown in Table 4. The total linear shrinkage of the height and diameter of LRC is generally low at a lower temperature. LRC exhibited the highest total linear shrinkage at 1000 °C, which is due to its lower $SiO_2$ content compared with the other two samples. This is also consistent with LRC having the lowest amount of free phase silica or quartz among the three samples based on the aforementioned results of the proximate analysis using the feldspar convention. Quartz plays a major role in the creation of a skeleton that minimizes shrinkage and deformation during the solidification phase [65]. The average loss on ignition of both LRC and KDE is lower than the mass loss obtained from TGA at 1000 °C (LRC: 16.74%, and KDE: 11.82%). SBC has a loss on ignition of 1.04%, which means that it will have a negligible effect on the mass loss when used in a clay-based material for ceramic production. The loss on ignition of the three samples are due to the removal of adsorbed and crystalline water and volatile materials.

**Table 4.** Physical properties of the samples studied as a function of firing temperature.

| Physical Properties | LRC | | SBC | | KDE | |
|---|---|---|---|---|---|---|
| | 1000 °C | 1200 °C | 1000 °C | 1200 °C | 1000 °C | 1200 °C |
| Total Linear Shrinkage, H | 8.69 ± 0.82 | 20.00 ± 1.28 | 1.03 ± 0.27 | — | 8.59 ± 0.65 | — |
| Total Linear Shrinkage, D | 8.85 ± 0.04 | 21.23 ± 0.04 | 0.62 ± 0.00 | — | 8.10 ± 0.18 | — |
| Loss on Ignition | 16.74 ± 0.71 | 17.37 ± 0.07 | 1.05 ± 0.17 | — | 11.82 ± 0.13 | — |
| Water Absorption | 24.69 ± 2.05 | 6.26 ± 0.08 | 19.63 ± 1.83 | — | 34.32 ± 2.47 | — |
| Apparent Porosity | 43.53 ± 1.79 | 11.19 ± 0.51 | 31.61 ± 3.12 | — | 43.64 ± 1.14 | — |

Note: "—" no data obtained, pellets were fused on the flat tiles.

At 1000 °C, KDE has the highest water absorption and apparent porosity of 34.32% and 43.64%, respectively, compared with the other two samples. These could be attributed to the porous structures in the KDE, where diatomite particles are still bound together without significantly changing the pore structure of the diatomite powder (Figure 7c). However, when the firing temperature is raised to 1200 °C, a melt phase covers the diatomite particles and fills the diatomite pores causing the KDE sample to fuse on the flat tiles as evident in Figure 9c. SBC has the lowest average water absorption of 19.63% and an apparent porosity of 31.16% at 1000 °C. When fired at 1200 °C, SBC exhibited a melting behavior resulting in its pellets sticking to the flat tile. This characteristic is similar to the sintering behavior of volcanic ash [66], which shows the potential of the SBC as a source of fluxes in ceramic production. LRC has an average water absorption of 24.69% at 1000 °C and 6.25% at 1200 °C. The decrease in water absorption and apparent porosity with increasing temperature is due to the vitrification phenomenon where liquid phases in the clay start to form and fill the matrix. These particles underwent rearrangement, which promotes the densification and contraction of the interior structure by the action of capillarity and surface tension [67]. Furthermore, the results of the water absorption of KDE, SBC and LRC were evaluated based on the international standard ISO 13006:2018 for ceramic tiles [68]. KDE, SBC and LRC fired at 1000 °C are potentially porous types of ceramic tiles as they pass standard water absorption requirements (>10%) for Group III (high water absorption).

After firing the samples at 1200 °C, both KDE and SBC are potential raw materials for the production of stoneware, since they passed the standard requirement of less than 3.0% for Group I (low water absorption) ceramic tiles, while LRC passed the standard requirement of 6 to 10% for semi-porous or Group II (medium water absorption) tiles.

The foregoing results could mean that Linamon red clay (LRC) could be a raw material for clay according to the ternary diagram [42,69] for various stoneware and ceramic structural applications in combination with other essential raw materials such as feldspar to reduce its water absorption and apparent porosity [19,70,71]. LRC could also be a raw material for terracotta production for pottery and bricks. Quartz could also be added to LRC to reduce its shrinkage [19,48,72].

Moreover, KDE was found to exhibit a plasticity suitable only for brick production. In addition to its plastic behavior during the forming stage of production, KDE was found to have fused parts when fired at 1200 °C, which is advantageous in reducing the porosity and water absorption of terracotta and structural ceramic wares [3,19,29,46,73]. This could mean that Kapatagan diatomaceous earth or KDE could be a secondary clay content, as well as a good source of flux for a certain firing temperature range. Furthermore, Salvador black cinder or SBC is a potential replacement or supplement for feldspar due to its flux potential. SBC is recommended for further investigation when mixed with other silicates for its optimum application.

## 4. Conclusions and Future Works

The three materials are found to exhibit unique physical and chemical properties essential for traditional and structural ceramic applications.

- Of the three raw materials, SBC has the highest cumulative passing size of 96.53% in a 150-micron sieve, which makes it advantageous for ceramic applications requiring raw materials of less than 150 microns.
- SBC is reported to be nonplastic, and LRC and KDE are classified as plastic materials with respect to their Atterberg limits and are suitable for brick production. In addition, LRC has potential for pottery production.
- The chemical compositions of the three samples reveal high silica, iron oxide and magnesium oxide. Only LRC is considered to have high alumina clay deposit levels due to its high alumina content. Their common mineral contents are montmorillonite, quartz and andesine. LRC is a potential source of clay for the slip casting production of ceramic tiles.
- Based on its thermal behavior, SBC exhibited a negligible effect on mass loss when used in clay-based materials for ceramic production.
- Black cinder of Salvador has the darkest shade of red and has the highest fusion coverage on the platform among the three materials at 1200 °C, which makes it a potential replacement or supplement for feldspar in clay-based triaxial materials for ceramic production.
- KDE could be used for secondary clay content or as a good source of flux for a certain temperature range.

Future research on these raw materials could include the evaluation of their physico-chemical properties and compressive strengths.

**Author Contributions:** Conceptualization, I.C.B.-A. and R.V.R.V.; methodology, I.C.B.-A., R.V.R.V., F.J.A.E., L.R.L., J.P.C., E.U.A.J., R.V.M.D., S.K.D.D., C.J.C.S., B.L.B. and A.M.R.S.; software, I.C.B.-A., F.J.A.E. and R.V.R.V.; validation, I.C.B.-A., R.V.R.V., F.J.A.E., L.R.L. and B.L.B.; formal analysis, I.C.B.-A., F.J.A.E., L.R.L., B.L.B., A.M.R.S. and R.V.R.V.; investigation, I.C.B.-A., F.J.A.E., L.R.L., J.P.C., E.U.A.J., R.V.M.D., S.K.D.D., C.J.C.S., B.L.B., A.M.R.S. and R.V.R.V.; resources, I.C.B.-A., F.J.A.E., L.R.L., J.P.C., E.U.A.J., R.V.M.D., S.K.D.D., C.J.C.S., B.L.B., A.M.R.S. and R.V.R.V.; data curation, I.C.B.-A., F.J.A.E., L.R.L., B.L.B., A.M.R.S. and R.V.R.V.; writing—original draft preparation, I.C.B.-A., F.J.A.E., L.R.L., B.L.B., A.M.R.S. and R.V.R.V.; writing—review and editing, I.C.B.-A., F.J.A.E., L.R.L., B.L.B., A.M.R.S. and R.V.R.V.; visualization, I.C.B.-A., F.J.A.E., L.R.L., B.L.B., A.M.R.S. and R.V.R.V.; supervision, I.C.B.-A., F.J.A.E., L.R.L. and R.V.R.V.; project administration, I.C.B.-A., R.V.R.V. and L.R.L.; funding acquisition, I.C.B.-A. and R.V.R.V. All authors have read and agreed to the published version of the manuscript.

**Funding:** This research was funded by the Department of Science and Technology Philippine Council for Industry, Energy, and Emerging Technology Research and Development (DOST-PCIEERD), under grant number 7374.

**Acknowledgments:** The authors acknowledge the following organizations: Ceramic Engineering and Department of Materials and Resources Engineering and Technology of Mindanao State University—Iligan Institute of Technology and the Department of Science and Technology Philippine Council for Industry, Energy, and Emerging Technology Research and Development (DOST-PCIEERD). Acknowledgement is also extended to John Louie L. Tefora for his administrative and laboratory assistance.

**Conflicts of Interest:** The authors declare no conflict of interest. The funders had no role in the design of the study; in the collection, analyses, or interpretation of data; in the writing of the manuscript; or in the decision to publish the results.

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
