# Peer review of "Evaluation of Linamon Red Clay, Salvador Black Cinder and Kapatagan Diatomaceous Earth of the Southern Philippines"

_minerals, doi:10.3390/min13020252_

Round 1

Reviewer 1 Report

In this manuscript the authors have studied silicate minerals from the southern island of the Philippines, including the province of Lanao del Norte. However, as they point out, some of these resources in the region have not been used as raw materials in the production of various ceramic products for industrial, pharmaceutical, and nanotechnology applications. These could include tiles, sanitary ware, dinnerware, insulating bricks, porcelain, membranes and coatings. Some of the explored minerals for this study include the red clay in the municipality of Linamon, diatomaceous earth in Kapatagan and black cinder in Salvador. It is the aim of this study that these minerals are evaluated on their physical and chemical properties so that these will be used for optimum application. The properties that were determined are their specific gravities, raw and fired surface colors, Atterberg limits, particle size distribution, thermal properties, morphologies and their mineralogical compositions. Pellets were formed for each raw material and fired at two temperature levels 1000°C and 1200°C to evaluate its physical properties. After sieving, the Linamon red clay has a 38.88% cumulative passing size of 150μm, while the black cinder of Salvador and diatomaceous earth of Kapatagan have cumulative passing sizes of 96.53% and 60.12%, respectively. The primary mineral content of the three samples are montmorillonite, quartz, and andesine. Black cinder fired at 1200°C has the darkest shade of red with a greasy quasi-submetallic luster. It attained the highest fusion coverage on the platform among the three materials which makes it potential to supplement or replace feldspar in clay-based triaxial materials for ceramic production. The diatomaceous earth has the potential to be a secondary clay content source and a good source of flux for a certain temperature range. Both the red clay and diatomaceous earth were classified as plastic materials which are suitable for brick tile production. These are just a few of the features of the clay minerals in the region which present suitable properties for application as raw materials at least in the production of ceramic tiles and hollow ceramic products.

            Overall, the authors have done an excellent job of first identifying the potential use of clay deposits in the Philippines and then providing appropriate laboratory data to make a convincing argument for their ceramic applications. I am pleased to recommend that it is acceptable for publication. However, there are a few, very minor grammatical corrections that are needed throughout the manuscript. So I recommend that someone skilled in English grammar be employed to review the text as the final version is being prepared.

Author Response

To Reviewer 1,

Reviewer 2 Report

The MS titled "Evaluation of Linamon red clay, Salvador black cinder and Kapatagan diatomaceous earth of Southern Philippines" can be accepted for publication after the authors address following major comments:

1.     Introduction section is weak. Please improve this section by including the analysis of previous studies. 

2.     Please include the significance, novelty, and objectives of this study in the last paragraph of introduction section. 

3. Figure 1 is not clear. Please replace it with high resolution image.  

4.     In the results and discussion section, the comparison with previous studies is weak. Please compare your results with following published articles (but not limited to). Also, you may use the following and other articles to further improve the introduction section:

a.     Dang, L. C., Khabbaz, H., & Ni, B. J. (2021). Improving engineering characteristics of expansive soils using industry waste as a sustainable application for reuse of bagasse ash. Transportation Geotechnics, 31, 100637.

b.     Shahsavani, S., Vakili, A. H., & Mokhberi, M. (2020). The effect of wetting and drying cycles on the swelling-shrinkage behavior of the expansive soils improved by nanosilica and industrial waste. Bulletin of Engineering Geology and the Environment, 79(9), 4765-4781.

c.     Danish, A., Totiç, E., Bayram, M., Sütçü, M., Gencel, O., Erdoğmuş, E., & Ozbakkaloglu, T. (2022). Assessment of Mineralogical Characteristics of Clays and the Effect of Waste Materials on Their Index Properties for the Production of Bricks. Materials, 15(24), 8908.

Please note that authors must compare every result obtained with previously published studies to show the consistency. 

5. Please use plasticity limit/plasticity index chart to show where this clay can be used?

6. Please include recommendations for future investigations. 

Author Response

To Reviewer 2,

Reviewer 3 Report

The study presents some valuable data on three distinct raw ceramic clays from the Philippines, intending to find a purpose for them and increase the ceramics production in that area, with a suitable price. However, there are serious remarks and this research needs to be placed in the appropriate framework to correspond to reality. There are many remarks, but the most important are as follows.

-        The aimed products are vaguely enumerated in all the sections. What can be seen from the obtained results, only terra cotta tiles could be produced of these materials. It is not clear what “brick tiles” are. You maybe mean terra cotta ceramic tiles. The results on water absorption should be compared to the relevant standards. -        The Introduction section explains that the imported ceramic tiles do not along with standards. Which standards, and in which behavior, parameter? -        The most important remark is an insufficient understanding of the tested materials and ceramic products, along with the XRD pattern analysis. It would be very useful if you can test the samples using XRD by an experienced mineralogist and compare the peaks obtained with certain PDF databases while removing the noise corresponding to the amorphous phase. Smectites (montmorillonite) belong to clay minerals that swell in drying and firing, so they would only be a burden for the usage of these materials in the ceramic industry. Other minerals “detected” can also be somewhat strange. XRD would certainly detect the carbonates in all the samples. As an example of a proper XRD analysis, chemical analysis and other parameters, you could consider this paper 10.3390/ma15093145. -        Chemical analysis is not properly done, since there is no loss on ignition presented, which would significantly change all the other oxides concentrations. The content of Al2O3 in non-plastic clay is low, and that is why the material is not plastic, which is related to the low clay mineral content. -        The content of calcium and magnesium carbonates is significant and could appear as Ca(OH)2 grains after the water absorption test, which is not discussed. -        SEM screening should be placed before thermal analysis. The appearance could be checked in the literature, especially for the specific structure of the KDE clay. -        The tested samples can be assigned as raw clays, but not clay minerals since they do not contain only clay minerals. -        There is no estimation of the size of the deposits. -        Fusion test is not a relevant expression, it is just termed as firing or sintering. -        The dark color of the raw SBC sample is possible of organic matter, which is not discussed. -        Why were the plates actually made? You should have performed firing at the refractory plate from the electric furnace. It seems not all plates were made of the same materials as the rollers, as claimed in the text.

Author Response

To the reviewer 3, 

Round 2

Reviewer 2 Report

The manuscript can be accepted as most of the comments are addressed. 

Reviewer 3 Report

The authors have significantly improved the paper. However, there are stil remarks and arguments about the results and conclusions.

-          XRF cannot detect loss on ignition, but the value is recoreded separatelly after firing at 900-1000 °C, and then the result is used to re-calculate the contents of major oxides. This is how the complete picture is obtained, and what is always done (check the new literature). Maybe the result obtained by TGA analysis may be used, you may check the literature.

-          Ca(OH)2 peaks could not be found in the XRD of the raw materials, since it is created after firing and moist absorption (from the air or after water absorption test, or in a chamber with aerated water). The peaks from carbonates are not observable from DTA diagrams, but the XRF analysis may be misleading here.

-          Many of the references are old, the article should consider newer ones.

-          Ref. 2 cannot be accessed through the link provided.

-Ref. 4 cannot be found, it seems it is not well referenced since it is non-existing on google.

ref. 16 also includes the pricing for the book which is also not accessible to me.

Other than yours and some of the not accessible books, no proof of usability of smectites in building ceramics is provided. Consider the provided XRD analysis, which claims the raw clays contain montmorillonite, saponite and nontronite, belonging to smectite clay minerals group. The point here is that the determination of XRD peaks might not be well done, since the smectites are well known to cause problems in shaping, drying and also firing of ceramic ware if present in high quantity. Since the XRD is done for the bulk sample, other techniques should be used, like separating the clay fraction and then analysing also the oriented preparations (you may see 10.1180/claymin.2017.052.4.04).

-          It is written (conclusion section) that all the 3 samples contain high content of alumina, which is not true.

Round 3

Reviewer 3 Report

I believe this paper can be published since the information on the clays from the Philippines is lacking in the literature, but please bear in mind my recommendations for future research.